# Hypersatellite K$_\alpha$ Production in Trapped Ar Ions at KK Trielectronic Recombination Energies



**Weronika Biela-Nowaczyk** [1,2,*], **Pedro Amaro** [3], **Filipe Grilo** [3], **David S. La Mantia** [4], **John Tanis** [4,†] **and Andrzej Warczak** [2]

1. GSI Helmholtzzentrum für Schwerionenforschung GmbH, 64291 Darmstadt, Germany
2. Institute of Physics, Jagiellonian University, 31-007 Krakow, Poland
3. Laboratory of Instrumentation, Biomedical Engineering and Radiation Physics (LIBPhys-UNL), Department of Physics, NOVA School of Science and Technology, NOVA University, 2829-516 Caparica, Portugal
4. Department of Physics, Western Michigan University, Kalamazoo, MI 49008, USA
* Correspondence: w.biela-nowaczyk@gsi.de
† Deceased.

**Abstract:** We report measurements of hypersatellite radiation of argon ions in the electron energy region of 5200 eV to 7500 eV. Here, we observed a strong enhancement of this hypersatellite K$_\alpha^h$ production. Trielectronic recombination (TR) is discussed as a possible channel for K$_\alpha^h$ production leading to this enhancement where main TR resonances are expected to occur. Data analysis was mainly based on the extracted intensity ratio of hypersatellite K$_\alpha^h$ to K$_\alpha$ lines (K$_\alpha^h$/K$_\alpha$). In addition, the collisional excitation and the collisional ionisation of the K-shell ions were modeled as main background processes of the K$_\alpha$ X-ray production. The K$_\alpha^h$/K$_\alpha$ intensity ratio shows a significant rise around 6500 eV electron energy by a factor of about two above the background level. This observation is compared with calculations of the expected electron energies for the resonant K$_\alpha^h$ emission due to the KK TR process. The observed rise as a function of the electron collision energy, which occurs in the vicinity of the predicted TR resonances, is significantly stronger and energetically much wider than the results of theoretical calculations for the TR process. However, the experimental evidence of this process is not definitive.

**Keywords:** electron beam ion trap; EBIT; multi-electron recombination processes; trielectronic recombination; flexible atomic code

## 1. Introduction

Electronic recombination is a fundamental process in electron–ion collisions. This process is present in all plasmas and is especially important in high-temperature plasmas produced by both astrophysical objects [1] and in the laboratory [2]. Among the recombination processes, dielectronic recombination (DR) is essential for both photon emission and ionic balance.

Recombination can result in non-resonant capture with the emission of a photon, called radiative recombination (RR) [3], or resonant capture with the excitation of one (or more) bound-state electron(s). In the case of DR, the excitation includes one electron. The first step of the DR process is called dielectronic capture and essentially is the time inverse of Auger decay. Following the resonant capture and the electron excitation, the ion can then decay to the ground state through radiative emission, which corresponds to the case of DR, or by the emission of an electron (Auger decay) [4]. An example of this process is shown in Figure 1a for the K → LL transition. The standard Auger notation is used throughout this paper: $A_i \to A_f B$, where the free electron is captured to the $B$ shell causing the promotion of a bound electron from the $A_i$ to the $A_f$ shell (left side of Figure 1a). This excited atom can then stabilize by photon emission (right side of Figure 1a).

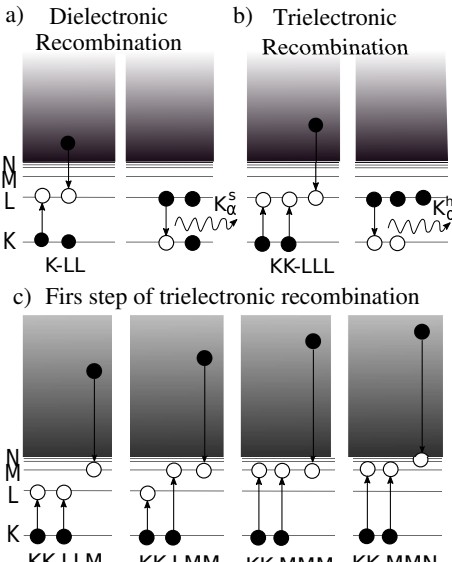

**Figure 1.** Energy schematic for examples of (**a**) K → LL dielectronic recombination followed by characteristic $K_\alpha^s$ emission, (**b**) KK → LLL trielectronic recombination followed by characteristic $K_\alpha^h$ emission and (**c**) examples of the first step of TR: KK → LLM, KK → LMM, KK → MMM and KK → MMN. The resonant electron energy is indicated by the vertical position of the free electron within the continuous energy states (gray area).

The DR process has been investigated extensively by theoreticians and experimentalists over recent decades using a variety of techniques, including merged electron and ion beams [5–11] and electron beam ion traps (EBITs) [2,12–18].

Trielectronic recombination (TR) is a process in which an ion–electron collision results in the resonant capture of a free electron to an ion with two core electrons being excited to higher atomic shells. Note that this definition follows a mono-configuration picture, however it is widely used in the community to define resonances with major contributions of two-core excitations.

These multi-electron transitions can refer to excitations within the same shell (intrashell transitions) or with one electron changing shell (intershell transition). Strong resonances of intrashell TR processes from Be-like chlorine [19] and silicon [20] have been observed at heavy ion storage rings. The intershell transitions are represented as KL-TR and have been investigated with an EBIT using Ar, Fe and Kr ions [21–25]. Results of intershell TR investigation at Jagiellonian University Electron Beam Ion Trap (UJ-EBIT) are reported in [26]. This present work is concentrated on the TR process where two electrons are excited from the K shell. An example of this process is shown in Figure 1b, where TR is shown for the KK → LLL transition. The first step of TR (Figure 1c) is a very rare process where a doubly excited ion state is produced. This process can be followed mainly by the emission of two sequential photons as the system stabilizes. A $K_\alpha$ transition to a hollow K-shell is termed hypersatellite emission ($K_\alpha^h$, see Figure 1b), while a $K_\alpha$ transition with a spectator K-shell electron is termed satellite emission ($K_\alpha^s$, see Figure 1a). To our knowledge, this multi-shell KK TR has not been a subject of study thus far.

In work that preceded the electron–ion TR measurements, it is noted that a similar process [27,28], referred to as resonant-transfer and double-excitation (RT2E), was investigated in ion–atom collisions, but without positive results. In Ref. [27], an upper limit to the RT2E cross section for He-like krypton ions was stated, but the value was outside the experimental limits of the apparatus.

In the present work, high-order resonances were investigated in the region of the KK → LLL, KK → LLM, KK → LMM, KK → MMM and KK → MMN transitions for a laboratory Ar plasma. The schemes of those five processes are presented in Figure 1b,c, where the rise in the resonance electron energy for this processes sequence is indicated.

The work here follows the observation of enhanced Ar $K_\alpha^h$ emission in an electron beam ion trap (EBIT) at electron energies ($E_e$) around 6.5 keV, which is indicative of a KK resonance associated with TR [29].

## 2. Experimental Setup

A compact, room-temperature (permanent magnet) EBIT was used to perform this experiment at Jagiellonian University (UJ). The UJ-EBIT is an ion irradiation facility model *S* produced by the commercial company DREEBIT https://www.dreebit-ibt.com/ion-sources.html (accessed on 1 March 2023). The trap schematic is shown in Figure 2. A heated cathode, with a diameter of 0.5 mm, in conjunction with a negative bias, supplies an electron beam to the trap. Permanent magnets in the trap (with magnetic induction of 250 mT on the axis) confine the electron beam to a radius of 25 μm [30,31]. The background pressure in the system was maintained at the level of $10^{-10}$ mbar, before a low-pressure, high-purity argon gas was introduced into the trap. The electron beam passed through the Ar gas cloud, thus creating Ar ions of various charge states through impact-induced ionization. The electron beam was then collected when leaving the trap (Figure 2). Typical cathode currents were close to 10 mA.

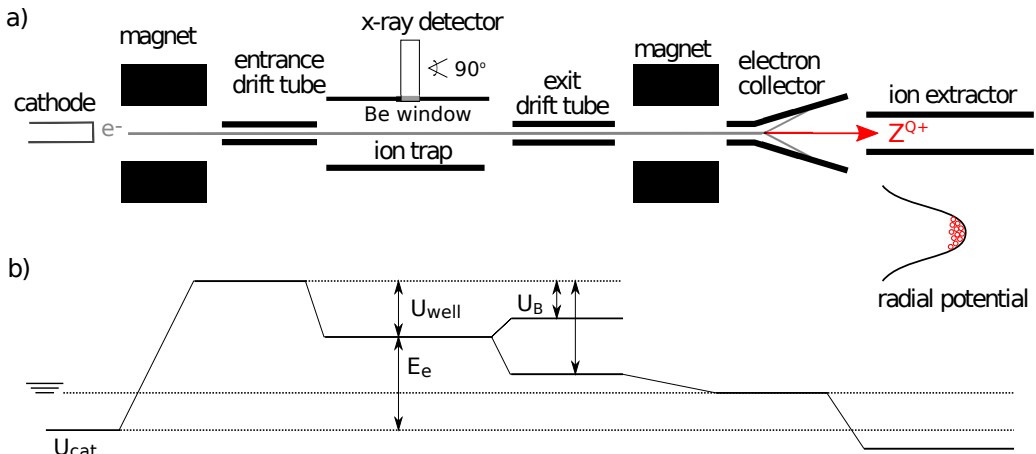

**Figure 2.** (**a**) Schematic of the Model S EBIT (DREEBIT). (**b**) Representation of all voltages at each electron beam stage.

Cylindrical drift tubes, comprising positively biased electrodes, confined the ions axially in a potential well. Radial trapping was achieved by the negative space charge density of the electron beam, which is enhanced by the radial beam confinement due to the magnetic field. The trap output capacity at an electron beam energy of 6400 eV is estimated as $2.5 \times 10^7$ positively charged particles. Evaporative cooling in the trap takes place principally in two ways. First, continually introducing neutral Ar directly into the trap creates low-$q$ ($q$—the positive charge, $q = 1, 2, \ldots$) ions with sufficient intensity, which allows them to carry away (while leaving continuously from the trap) a significant fraction of kinetic energy from the mid- and high-$q$ ions. Thereby, the ion thermal equilibrium for high-$q$ is achieved [32]. Second, lowering the "closed" potential ($U_B$, Figure 2) by a small amount (of a few eV) enhances evaporative cooling by allowing low-$q$ ions to escape more easily from the trap. Therefore, the concentration of highly charged ions in the ion plasma is further increased [17].

A silicon-drift detector (SSD) (XFlash Bruker 5030) with an energy resolution of 127 eV (FWHM) at the Mn $K_\alpha$ line and an active area of 30 mm$^2$ was positioned at 90° to the beam axis at the trap center to collect emitted X-rays. The trap and detector had combined Be windows of 25 μm, providing an efficiency of about 90% at the energy of characteristic Ar K X-rays. Furthermore, our acquisition system based on the TERX Detection System (https://www.dreebit-ibt.com/shop/terx-detection-system.html, accessed on 1 March 2023) resolves each X-ray event with a time resolution of 1 ms.

A typical time-integrated X-ray spectrum collected by the detector is shown in Figure 3 (time-resolved spectrum is shown in Section 3.2). As seen in the spectrum, several characteristic emission lines from elements present in the cathode (Ir) and physical materials of the experimental apparatus (Cr, Fe, Ni-components of stainless steel) are seen. Most prominent characteristic X-ray lines, in particular from singly-ionized Si detector atoms and stainless steel components, were used for X-ray energy calibration. The characteristic target Ar X-rays necessarily sit on top of an X-ray continuum background generated principally by the electron–nucleus interaction (bremsstrahlung [33]) of the electron beam with the edge at the electron beam energy ($E_e$ = 7640 eV for Figure 3). This significant intensity of the bremsstrahlung is mainly caused by the electrons lost from the electron beam which interact with the drift tubes. During the experiment, the fraction of lost beam was kept below $5 \times 10^{-3}$. The procedure of background subtraction is explained in detail in Section 3.4.

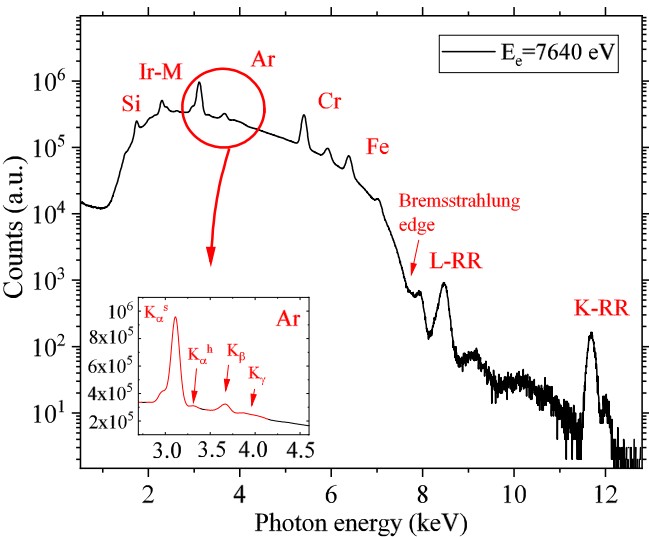

**Figure 3.** A typical spectrum collected with the UJ-EBIT apparatus. The inset shows details of the characteristic argon K line, the data shown in black were used for background subtraction.

## 3. Results and Discussion

### 3.1. Selection of the Expected TR Energy Region

The TR process, similarly to DR [34], is considered as a two-step process: $|i\rangle \rightarrow |d\rangle \rightarrow (|f\rangle + \hbar\omega)$. The theoretical position of resonances, their widths, cross sections as well as resonance strengths [26,35] were calculated with the Flexible Atomic Code (FAC) [36]. Here, the calculations focused on five different KK processes: KK → LLL, KK → LLM, KK → LMM, KK → MMM and KK → MMN. For comparison, calculations were also performed for the K→LL DR process. Moreover, in order to test the quality of the applied code, similar DR calculations for the Kr charge states were performed, which achieved an agreement with the results presented in Ref. [37]. In TR processes, the M shell is involved $K_\alpha^h$, $K_\beta^h$ and $L_\alpha$ lines can be produced. In the present experiment, only the $K_\alpha^h$ line is clearly seen and separated from other lines of the spectrum.

In most cases, He- and Li-like argon ions ($Ar^{16+}$ and $Ar^{15+}$) form the dominant fraction of the charge states of the EBIT plasma under the conditions used in the present experiment (Section 3.2). Therefore, as an example, the resonance strengths and cross sections calculated with FAC for the He-like and Li-like ion charge state are presented in Figure 4. Here, only the resonances which cause the $K_\alpha^h$ emission are shown. Note that, for the de-excitation of the intermediate state, the Auger effect has a considerable contribution. This fact was taken into account during the calculations. Based on Figure 4, one can conclude that the calculated positions of the strongest TR resonances correspond to electron energies of about 5500 eV, 6000 eV and 6500 eV for the KK → LLL, KK → LLM and KK → LMM resonances, respectively. The plotted cross sections were convoluted with additional broadening of

30 eV, which is typical for EBIT conditions. One can conclude that the total (convoluted) cross sections of TR are about a factor of 10,000 smaller than the respective of DR (Figure 4). An important observation from Figure 4 is an expected enhancement of $K_\alpha^h$ emission for the KK → MMN resonance at $E_e$ of about 7250 eV. One has to note that for the KK → MMM case, the $K_\alpha^h$ does not appear. Additionally, among the Auger de-excitation channels of the intermediate state, some of them can still produce doubly excited K-shell states. This process, called resonant excitation [18], can also result in $K_\alpha^h$ production. The positions of these resonances are in the vicinity of KK TR resonances. The calculations of KK → LMn resonant excitation (KK RE), performed for He-like Ar ions with $n$ = 3–9 showed that this way the $K_\alpha^h$ production is substantially enhanced in an electron energy range 6750–7000 eV. The KK RE processes involving higher atomic shells (e.g., KK → MMn, n ≥ 3) could also be responsible for $K_\alpha^h$ production at higher electron energies (above 7000 eV). However, for lower resonant electron energies (below 6750 eV), the KK TR process has been calculated to be about two orders of magnitude stronger than KK RE.

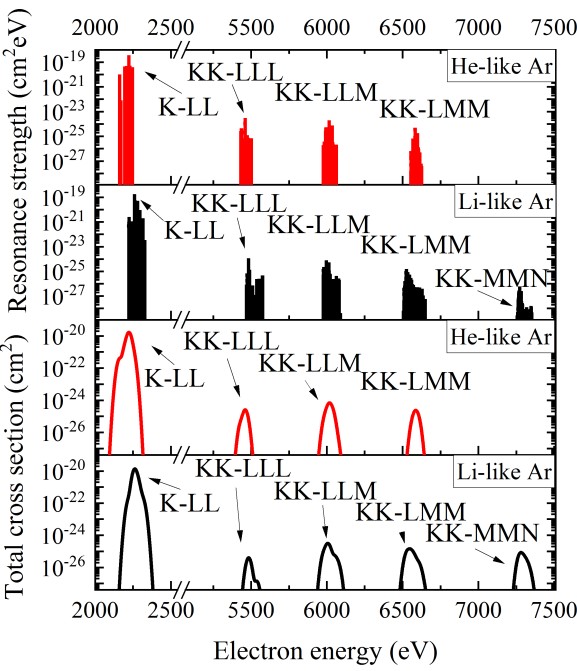

**Figure 4.** The resonance strengths and total cross sections of TR with the $K_\alpha^h$ emission for He-like and Li-like argon ions calculated with the use of the FAC [36]. Different signatures KK → LLL, KK → LLM, KK → LMM, KK → MMM and KK → MMN were calculated, results for $K_\alpha^h$ de-excitation are plotted and labeled. For comparison, results for the DR K → LL are also shown. Note the logarithmic scale of the resonance strength and cross section.

### 3.2. Ion Charge State Evolution in the Investigated Electron Energy Region

Paramount to the understanding of the physics within the trap is the ion charge state composition of the plasma. The trap was flushed periodically to control the production of highly charged ions. To verify that the distribution of charge states within the trap tends towards highly charged ions as trapping time evolves, radiative decay rates and corresponding photon energies were calculated. FAC calculations included $K_\alpha$ transitions in Ar ions from excited states with one K-shell vacancy to final states (both ground and excited) as well as the transitions from excited states with two vacancies in the K-shell (hypersatellite). The energies of $K_\alpha$ and $K_\alpha^h$ transitions were analyzed as a function of decay rates. These functions showed an asymmetric shape and were then used to calculate an average value weighted by the decay rates for the $K_\alpha$ photon energy (Figure 5). Because of the asymmetry of these shapes, bigaussian functions have been fitted and used to estimate the error bars.

The FWHM for these asymmetric distributions, divided in two parts, we present as asymmetric error bars plotted in Figure 5. In this Figure, the $K_\alpha$ and the hypersatellite $K_\alpha^h$ energies for Ar ions from $q = 6+$ to $q = 17+$ are shown. The calculations for ions with charge states lower than 6+ were not carried out, because for these charge states the average $K_\alpha$ photon energy decreases by less than 10 eV only. Moreover, based on the results of the calculations presented in Figure 5, the $K_\alpha$ line position stabilizes at the charge states where the L-shell is completely filled. The absolute number of loosely bound M shell electrons is less relevant.

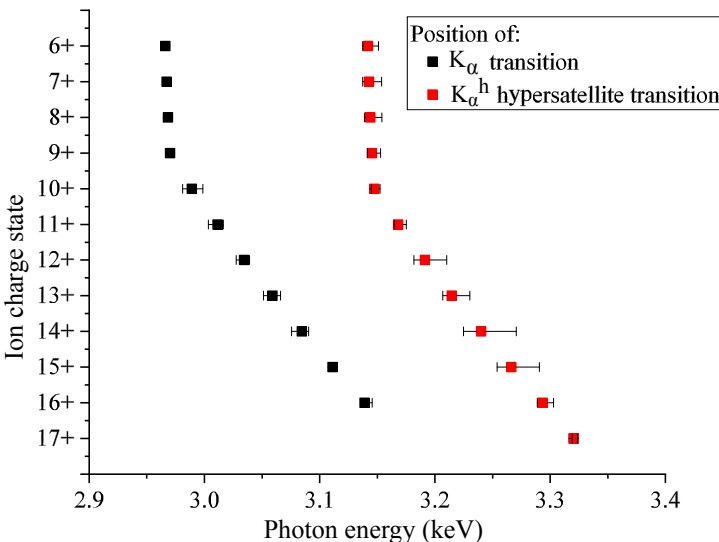

**Figure 5.** Averaged values of $K_\alpha$ and $K_\alpha^h$ photon energies for various Ar charge states calculated with FAC. For $q = 17+$, only hypersatellite transitions are possible.

For illustration of the charge state evolution, an X-ray spectrum recorded over a total trapping time of 500 ms is shown in the left part of Figure 6. Here, the position of the $K_\alpha$ line is clearly changing with rising ionisation time and the shift to higher energy is seen. These data were divided into three trapping time intervals and subsequently corrected for the detection efficiency. The background was subtracted as well. In order to reveal the distribution of charge states of Ar in the ion plasma, Gaussian X-ray line profiles, centered at the $K_\alpha$ line positions shown in Figure 5, were fitted. The sums of these profiles match the total experimental X-ray spectra well. Results of this procedure for the three trapping time intervals are shown in the right-hand side of Figure 6. In the first 100 ms of the trapping time (bottom panel), almost all charge states are present in the plasma. Here, the X-ray radiation from the mid-$q$ ions ($q \approx 10$) dominates and the overall emission profile is centered close to 3.03 keV (see bottom of Figure 6). For the next 101–250 ms trapping time (middle panel), the population of B-like to He-like ions begins to rise and the contribution of low-$q$ ions ($\leq 10$) falls below 5%. Therefore, the center position of the total emission shifts to higher energies by about 75 eV up to 3.10 keV. Above 250 ms trapping time (top panel of Figure 6), emission from highly charged ions begins to dominate. The low-energy tail (below 2.95 keV) of the total emission peak almost disappears and the main portion of the total emission peak narrows. Here, the radiation from the He-, Li-, and Be-like ions strongly dominate. This illustrates that, for the trap parameters and electron beam energies selected for the experiment, the radiation from $Ar^{13+}$ - $Ar^{16+}$ ions dominates over radiation from the low-charged ions (LCI). One has to note that the distribution shown in Figure 6 is not a charge state distribution but a distribution of contribution of particular charge states to the $K_\alpha$ line emission. In order to reveal the real charge state distribution, cross sections for $K_\alpha$ production for different ion charge states should be considered.

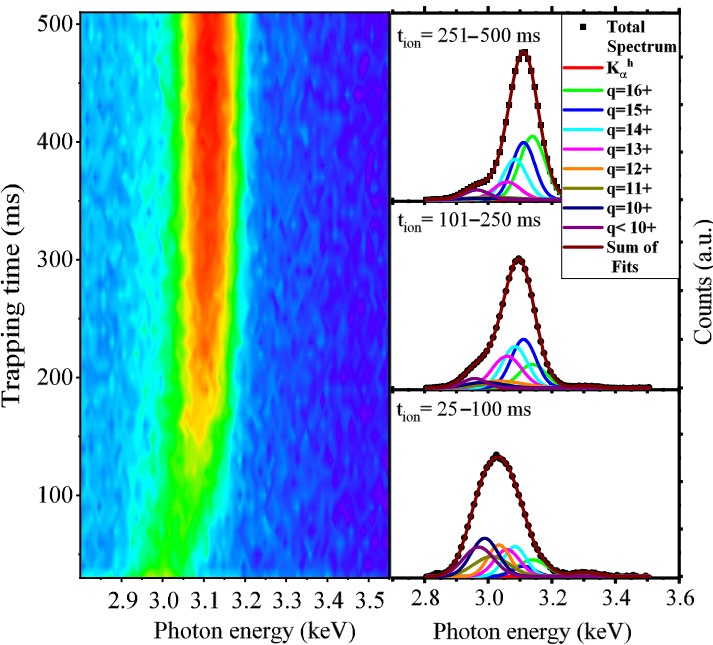

**Figure 6.** Data collection for $E_e$ = 6440 eV and an Ar gas pressure of $2.5 \times 10^{-10}$ mbar. Here, the charge-state-dependent $K_\alpha$-line intensity evolution is presented as a function of trapping time ($t_{ion}$). The plots on the right correspond to $K_\alpha$ accumulated spectra for various time windows.

The experimental approach applied within this work is based on the analysis of the hypersatellite $K_\alpha^h$ transition. Results of the calculations of the corresponding transition energies are shown in Figure 5. The position of the $K_\alpha^h$ energy varies similarly to that for $K_\alpha$, with the $K_\alpha^h$ energy generally significantly higher than the $K_\alpha$ energy. In the case of the highest ion charge states (higher than $Ar^{12+}$), the $K_\alpha^h$ energy exceeds 3.2 keV. However, for ion charge states lower than $q = 10+$ this energy is very close to the $K_\alpha$ transition energy of high-charge states (in particular to $K_\alpha$ of $Ar^{16+}$). Based on the calculations shown in Figure 5, it has been concluded that

1.  Even if the TR process would take place for lower charge states, our measuring method would not be sensitive enough to this process. Thus, the collected hypersatellite $K_\alpha^h$ radiation would be only a very small fraction ($\approx 10^{-4}$) of a much stronger $K_\alpha$ line.
2.  The observed $K_\alpha^h$ X-ray radiation is well separated from the $K_\alpha$ background for the highly charged ions.

### 3.3. Simulations of the Time Evolution of the Ion Charge State

As previously discussed in Section 3.2, the ion charge state evolution is crucial for explanation of the X-ray line profiles. Therefore, simulations were performed to gain a deeper understanding of the time evolution of the ion charge state distribution for different electron energies.

The simulations were based on the numerical solving of a set of nineteen coupled differential equations, one for every charge state (including the bare ion). Those differential equations are as follows:

$$
\begin{aligned}
\frac{dN_q}{dt} =& n_e v_e \left[ N_{q-1}\left( \sigma_{q-1}^{CI} + \sigma_{q-1}^{EA} \right) + N_{q+1}\sigma_{q+1}^{RR} \right. \\
& \left. - N_q\left( \sigma_q^{CI} + \sigma_q^{EA} \right) - N_q\sigma_q^{RR} \right] \\
& - N_0 N_q \sigma_q^{CX} \overline{v}_q + N_0 N_{q+1}\sigma_{q+1}^{CX}\overline{v}_{q+1},
\end{aligned}
\tag{1}
$$

where $N_q(t)$ represents the population of a charge state $q$, $N_0$ was estimated based on the Ar gas pressure in the trap, $\sigma$ denotes the cross section of an ionizing or recombination process, and $n_e$ and $v_e$ represent the electron density and velocity, respectively. These equations

take into account collisional ionization (CI), excitation and subsequent autoionization (EA), radiative recombination (RR) and charge exchange (CX) with the background residual gas. The presented simulations were previously benchmarked by modeling a time-dependant X-ray spectrum of highly charged Fe ions [35].

As presented in the introduction, the resonant production of the $K_\alpha^h$ line is a signature of the proposed TR process. Therefore, the data analysis (presented below) is based on the observation of the intensity variation of the $K_\alpha^h$ line for different electron energies used in the EBIT. In this observation, it is important to estimate the evolution of the background processes in which the $K_\alpha^h$ line can be produced. Here, the main background process causing the $K_\alpha^h$ emission is collisional excitation (CE) of the H-like ions [36]. The intensity of this process depends not only on the cross section $\sigma_{17+}^{CE}$, but also on the density of the H-like ions in the mixture of ions in different charge states. The resonant excitation [18] of H-like ions was also considered, however the main channels (Kmn, with m, $n = 3,4, \ldots$) are located around 3 keV to 4 keV. Hence, this process does not contribute to the background in the observed region.

Results of the simulation for trapping times in the range 100–250 ms and for gas pressure $2.5 \times 10^{-10}$ mbar are presented in Figure 7. A steady growth of the H-like population with the electron beam energy was found. Ly-$\alpha$ emission induced by the CE of the H-like ions is the main source of background for the TR process. Consequently, this $K_\alpha^h$ background emission increases with the electron beam energy.

To model the experimental results discussed in Section 3.4, it is necessary to calculate the behavior of the $K_\alpha^h/K_\alpha$ intensity ratio (R). Therefore, the simulations of the $K_\alpha$ background production consider emissions induced by CE of the K-shell for charge states $q = 1+, \ldots, 16+$, as well as emission induced by CI of the K-shell for charge states $q = 0, \ldots, 15+$. The background ratio is then given by

$$R = \frac{\sigma_{17}^{CE} \int_{t_0}^{t1} N_{17+}(t)dt}{\sum_{q \geq 1+}^{16+} \sigma_q^{CE} \int_{t_0}^{t1} N_q(t)dt + \sum_{q \geq 0}^{15+} \sigma_q^{CI} \int_{t_0}^{t1} N_q(t)dt}, \qquad (2)$$

where $t_0$ and $t_1$ are the limits of the respective time windows.

Equation (2) shows that this background depends on the charge state distribution of the ion plasma in the investigated trapping time window. R simulations were performed for two time windows (100–250 ms and 250–500 ms presented in Figure 6) and will be compared with experimental data below in Section 3.5.

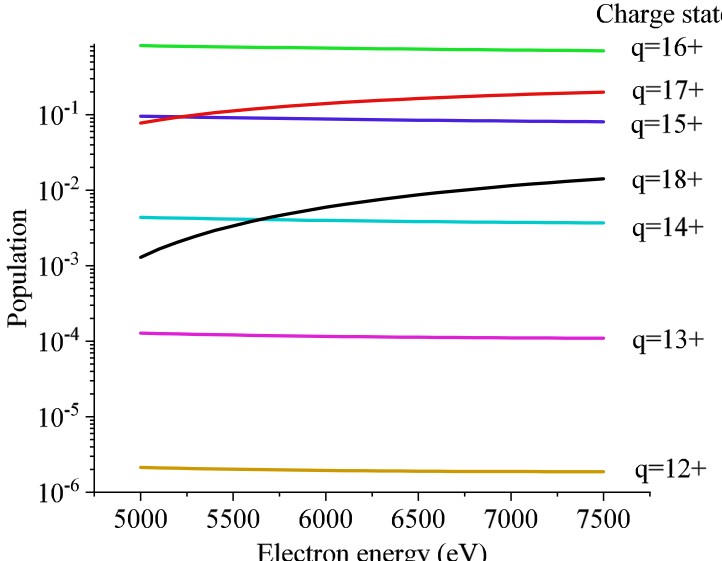

**Figure 7.** Simulated evolution of the charge state populations. Here, the trapping time was 100–250 ms with a gas density of $2.5 \times 10^{-10}$ mbar. Note that the $Y$ axis is logarithmic.

*3.4. Data Analysis*

Data presented in this paper were collected in two different measuring runs. During the first one, the data collection was carried out with the Ar gas pressure set at $2.5 \times 10^{-10}$ mbar. Data were gathered with X-rays events being sorted with trapping time, as in Figure 6. Here, the selection of an appropriate trapping time window is necessary for the experiment. As discussed already in Section 3.2 (Figure 5), $K_\alpha^h$ and $K_\alpha$ can be resolved only for the highest charge states, which is needed for TR observation. These charge states can be reached in trapping time windows 101–250 ms and 251–500 ms, as shown in Figure 6. However, one has to point out, additionally, that the presence of both K-shell electrons is required for any KK TR resonant transition (KK$\rightarrow |f\rangle$). So, it is necessary to minimize the fractions of $Ar^{17+,18+}$ ions in the trap. Therefore, the trapping time window 101–250 ms was chosen for the experiment. In this case, Li-like argon ions ($Ar^{15+}$) dominate. Note that this discussion is only valid for the gas pressure of $2.5 \times 10^{-10}$ mbar.

Data collection was carried out for 12 different electron energies with each point measured for about 70 h. For each run of data collection, the same X-ray range (2.8–5.0 keV) was chosen. In the X-ray window selected, short photon energy intervals, which are far from peaks (at the distance of three times FWHM, see Figure 3), were attributed to the background. Finally, the background was obtained based on the interpolation of the intervals prepared as above. Then, a multipeak fit was applied to the background-subtracted data. An example is shown in the inset of Figure 8. Here, the line profiles for the $K_\alpha$ transitions in the low argon charge states ($K_\alpha^{LCI}$ for $q = 0, \ldots, 11+$), the $K_\alpha$ transitions in the high argon charge states ($K_\alpha^{HCI}$ for $q \geq 12+$) and the hypersatellite transitions ($K_\alpha^h$) are resolved. This procedure was possible due to the good X-ray energy resolution of the detector. Moreover, an explanation of the separation of the $K_\alpha^{LCI}$ and the $K_\alpha^{HCI}$ line profiles is given on Figure 5 and the charge state evolution is presented in Figure 6. There, the $K_\alpha^{LCI}$ line is present in the spectrum from the very beginning of the trapping time, whereas the $K_\alpha^{HCI}$ line emerges only after about 100 ms.

Extreme care was taken in order to keep all trap parameters stable for the entire experimental effort. However, the electron density and formation of the electron beam (radial trapping potential) may slightly fluctuate. Radiative recombination to the argon K-shell was analyzed in order to normalize data to compensate these possible variations caused by changes in the trap working conditions. RR is a well-known process as well as having a smooth dependency with $E_e$. Theoretical calculations of the RR cross sections were performed with the Stobbe equation [38] and with FAC, with good agreement between both methods. The observed RR intensities and calculated RR cross sections were used for normalising the observed $K_\alpha^h$ intensities relative to the first measurement point ($E_e^1 = 5230$ eV), i.e., according to

$$I_n^h \text{ norm} = I_n^h \, \frac{I_1^{RR}}{I_n^{RR}} \, \frac{\sigma_{RR}(E_e^n)}{\sigma_{RR}(E_e^1)}, \tag{3}$$

where $I_n^h$ and $I_n^h \text{ norm}$ are the $K_\alpha^h$ total peak counts before and after normalization. $I_n^{RR}$ is the RR total peak count for the $n$th measurement for $Ar^{17+}$. $\sigma_{RR}(E_e^n)$ is the RR cross section for the $E_e^n$ energy. $I_n^h \text{ norm}$ takes into account fluctuations due to the measurement time $t_t$, the density of the electron beam $\rho_e$ and the ion population $N_q$, by noting that the counts $I$ of a given peak follow

$$I \propto t_t \, \rho_e \, N_q \, \sqrt{E_e} \, \sigma, \tag{4}$$

where $\sigma$ is the cross section of an atomic process leading to the emission. In this way, the normalised $K_\alpha^h$ counts were obtained and are presented in Figure 8. Additionally, an estimation of the $K_\alpha^h$ background due to only CE can be obtained from Equation (4) applied to the $K_\alpha^h$,

$$I_n^h \text{ back} = A \, N_{17+} \, \sigma_{CE}^{17+}(E_e^n) \, \sqrt{E_e^n}. \tag{5}$$

The factor $A$ is obtained when assuming the coincidence between $I_n^h$ $^{back}$ and $I_n^h$ for $n = 1$. $N_{17+}$ is obtained from the simulation shown in Figure 7. The result of Equation (5) is shown in Figure 8.

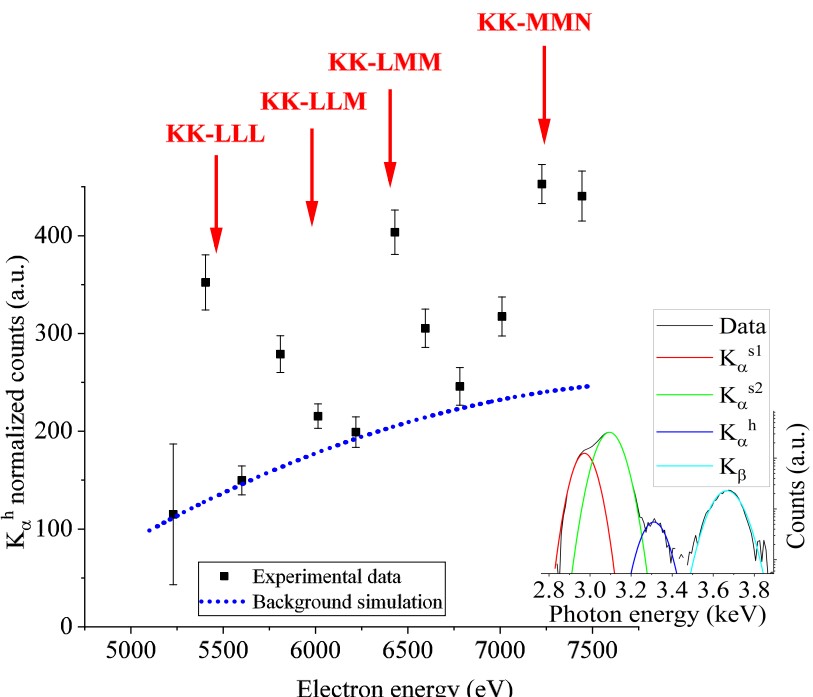

**Figure 8.** The normalized Ar $K_\alpha^h$ ($I_n^h$ $^{norm}$) variation scanned in the $E_e$ range expected for KK $\rightarrow$ LLL, KK $\rightarrow$ LLM, KK $\rightarrow$ LMM and KK $\rightarrow$ MMN TR. Here, the trapping time window of 100–250 ms and an Ar gas pressure of $2.5 \times 10^{-10}$ mbar were used. Red arrows indicate the theoretically calculated positions of the TR resonances. An example of an X-ray line fit is shown in the inset for an electron energy of 6430 eV (logarithmic scale).

*3.5. Observed Enhancement of the Hypersatellite Transition*

Figure 8 shows the $I_n^h$ $^{norm}(E_e^n)$ based on the procedure explained in Section 3.4. Here, one can observe a clean enhancement of the $K_\alpha^h$ production in the $E_e$ region expected from the TR calculations. The uniqueness of the KK TR process concerns the production of a state in the ion with two vacancies in the K-shell. Therefore, the main signature of this process manifests as a variation of the $K_\alpha^h$ radiation intensity with the electron energy. Based on the background simulations performed with the FAC, the $K_\alpha^h$ intensity should increase slightly. However, this is not the case for the experimental data presented in Figure 8. Instead, this figure presents much stronger resonant production of the $K_\alpha^h$.

In addition, to determine the reliability of this conclusion, the $K_\alpha^h$ and the $K_\alpha$ intensity ratios will be examined. The ratios are also independent of fluctuations caused by random variations of the EBIT working parameters and will be used to quantify the overall effect.

In Figure 9, the intensity ratios of $K_\alpha$ hypersatellite to $K_\alpha$ lines are shown as functions of $E_e$. The $K_\alpha^{LCI}$ and $K_\alpha^{HCI}$ line profiles have already been discussed above in Section 3.4. One has to keep in mind that the double structure of the $K_\alpha$ line (inset Figure 8) is due to the contribution of two groups of ion charge states: low-charge states (denoted by $K_\alpha^{LCI}$) and high-charge states (denoted by $K_\alpha^{HCI}$). However, as presented in the inset of Figure 8, the energy of the observed hypersatellite transitions is in the region of 3.2–3.4 keV. Figure 5 shows that this radiation is produced mainly by ions in the highest charge states ($q = 12$, ..., 17). Hence, as discussed already in Section 3.2, it is justified to focus the analysis on the ratio of the $K_\alpha^h$ and the $K_\alpha^{HCI}$ component of the $K_\alpha$ line (black points in Figure 9). The experimental results are compared with the background ratio (R), labeled as the simulation, given in blue in Figure 9. Its calculations were based on Equation (2) for the actual EBIT settings and with a 10% variation of the CI and RR cross sections for uncertainty estimation.

It is assumed that the charge states in the denominator of Equation (2) include only ions for $q \geq 12$. Therefore, the simulated R value represents the background for the $K_\alpha^h / K_\alpha^{HCI}$ intensity ratio.

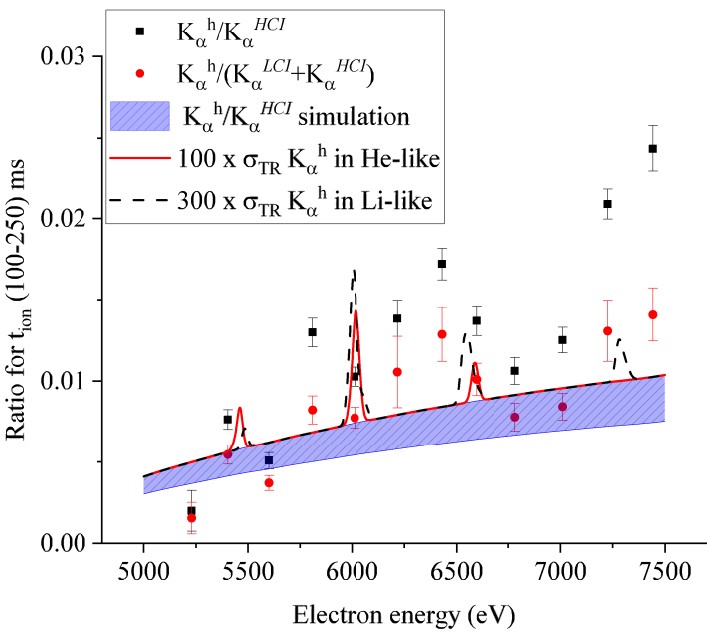

**Figure 9.** The argon $K_\alpha^h / (K_\alpha^{LCI} + K_\alpha^{HCI})$ and $K_\alpha^h / K_\alpha^{HCI}$ intensity ratios scanned for $E_e$ = 5200–7500 eV, $t_{ion}$ in the window 100–250 ms, Ar gas pressure of $2.5 \times 10^{-10}$ mbar. The simulation shows the background ratio (in blue) based on Equation (2) for $q \geq 12$, i.e., for $K_\alpha^h / K_\alpha^{HCI}$. In addition, the contribution of the KK TR processes ($\sigma_{TR}$) to the background ratio is demonstrated, for details see the text and labels in inset.

The data presented in Figure 9 show a similar pattern of $K_\alpha^h$ production like the one presented in Figure 8. In particular, at positions of KK → LLL, KK → LLM, KK → LMM and KK → MMN TR resonances, visible structures are present. The strongest enhancement of the $K_\alpha^h / K_\alpha^{HCI}$ ratio is observed for the $E_e$ around 6500 eV. The enhancement factor between the experimental $K_\alpha^h / K_\alpha^{HCI}$ values obtained from the fit and the simulated background level is $2.4 \pm 0.4$. The position of this maximum, close to 6500 eV, suggests that the KK → LMM TR process may be responsible for this effect.

Moreover, in Figure 9, results of the FAC calculations of the KK TR resonance profiles for He- and Li-like ions are superimposed on the background level R. In order to obtain a clear view of the contribution of the calculated cross sections of the TR processes to the background ratio R (Equation (2)), cross sections were multiplied by a factor of 100 and 300 for He- and Li-like ions, respectively.

In addition, two data points of Figure 9 for $E_e$ = 7225 eV and 7443 eV show a strong enhancement of the $K_\alpha^h / K_\alpha^{HCI}$ ratio value above the background ratio. The position of this enhancement (Figure 4) may suggests a presence of the KK → MMN TR resonance. However, in this region the KK RE can additionally be responsible for the $K_\alpha^h$ production (see Section 3.1).

In order to provide further evidence for the structures observed in the previous experimental run, additional measurements were performed. The variation of the $K_\alpha^h$ intensity in the selected electron energy region was examined for slightly changed trap conditions. The Ar gas pressure was set at the level of $1.5 \times 10^{-10}$ mbar and the trapping time window was shifted to 250–500 ms. For these particular parameters, some modification of the ion charge state distribution in the trap was expected to be obtained. Both a lower gas pressure and a longer trapping time result in the enhancement of the production of higher charge states. Again, 12 data points were measured in the $E_e$ range of 5200–7500 eV with each data point collected for about 70 h. Analysis for this measurement was based

on the variation of the $K_\alpha^h/K_\alpha^{HCI}$ intensity ratio, similar to the procedure applied for the previous run. Results of the analysis are presented in Figure 10. The simulation of the ion charge state distribution was performed again for these new ion trap conditions. The simulated background ratio level (based on Equation (2)), as well as the KK TR cross section contribution, are also presented in Figure 10.

Similarly to Figures 8 and 9, the experimental data shown in Figure 10 present the enhancement of the $K_\alpha^h$ production at $E_e$ positions expected for KK → LLL, KK → LLM, KK → LMM and KK → MMN TR resonances. Again, the strongest enhancement of the $K_\alpha^h$ production is seen for $E_e$ around 6500 eV (Figure 10). The enhancement factor between the experimental $K_\alpha^h/K_\alpha^{HCI}$ intensity ratio and the simulated background level is about $2.7 \pm 0.4$. These results support the conclusion that TR resonances can be responsible for the structures observed in Figures 8–10. It is noted that the position of the strongest enhancement of the $K_\alpha^h/K_\alpha^{HCI}$ intensity ratio in Figure 10 is slightly shifted to a higher $E_e$ value by about 150 eV if compared to Figure 9. However, for the strongest KK → LMM resonance, it is still in the electron energy range where it is expected to play a crucial role. Nevertheless, the position shift can be explained by a stronger contribution of KK → LMM TR in He-like than Li-like ions (see Figure 4). As already mentioned above, this change in charge state distribution was expected due to the application of a lower gas pressure and of a longer trapping time (Figure 6).

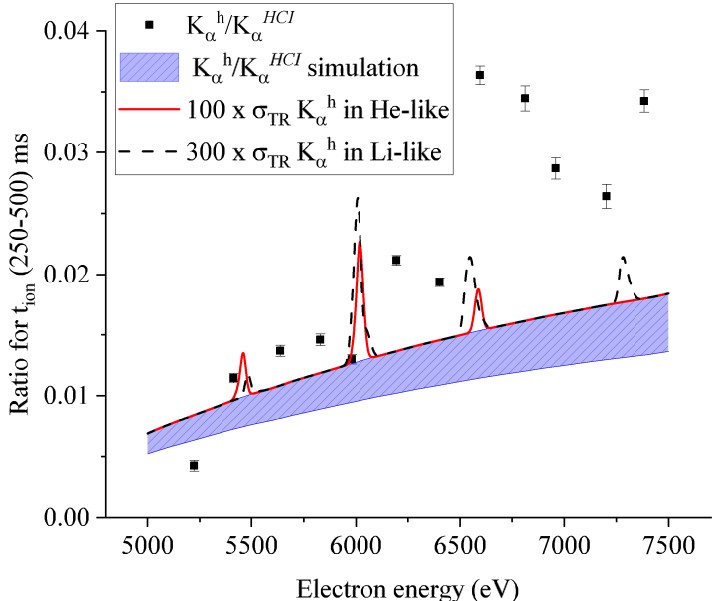

**Figure 10.** The argon $K_\alpha^h/K_\alpha^{HCI}$ intensity ratio scanned for $E_e$ = 5200–7500 eV, $t_{ion}$ in the window 250–500 ms, Ar gas pressure of $1.5 \times 10^{-10}$ mbar. The simulation shows the background ratio (in blue) based on Equation (2) for $q \geq 12$. In addition, the contribution of the KK TR processes ($\sigma_{TR}$) to the background ratio is demonstrated, for details see the text.

## 4. Conclusions

This paper presents a strong enhancement of the $K_\alpha^h$ production observed with the UJ-EBIT for argon ions in the electron energy range 5200–7200 eV. This resonant-like behaviour for particular electron energies suggests the presence of the multi-shell KK TR process. The strongest maximum is observed at the energy of about 6500 eV (an enhancement by a factor of about two) which suggests that the TR KK → LMM transition should be considered as the most effective among other KK-TR processes. This KK → LMM resonance was already discussed [29] to have the dominating impact on hypersatellite $K_\alpha$ production. It should be emphasized, however, that besides the KK → LMM TR transition, the KK → LLL, the KK → LLM and the KK → MMN TR processes (Figures 9 and 10) may also influence the presented results. The positions of these structures observed in the $K_\alpha^h$ production are compared with

the calculated positions of the KK TR resonances. This comparison gave us the first hint that the KK multishell TR process may be responsible, at least partially, for the observed structures. However, significant discrepancies between experimentally determined and calculated widths and intensities suggest that the experimental evidence of this process is not definitive. If the multishell TR process is the cause of the strong hypersatellite $K_\alpha^h$ production, it is much stronger and energetically wider than TR theoretical predictions. In addition, it was suggested that the KK RE process may substantially contribute to the $K_\alpha^h$ production for electron energies above 7000 eV.

To confirm these suggestions, the individual KK-TR resonances must be mapped out with the use of a much finer energy grid. Therefore, results of the present work deliver ideas for further experiments running on heavy-ion accelerators.

**Author Contributions:** Conceptualization, A.W. and W.B.-N.; investigation, W.B.-N., D.S.L.M., J.T. and A.W.; data curation W.B.-N., formal analysis W.B.-N., P.A., F.G. and D.S.L.M.; writing—original draft preparation W.B.-N.; writing—review and editing P.A., F.G., D.S.L.M., J.T. and A.W.; visualisation W.B.-N., P.A. and F.G.; supervision J.T. and A.W. All authors have read and agreed to the published version of the manuscript.

**Funding:** W.B.-N. acknowledges DSC Grant MNiSzW Nr 7150/E-338/M/2018, the GET_INvolved Programme at FAIR/GSI (www.fair-center.eu/get_involved) and the JIPhD program through contract POWR.03.05.00-00-Z309/17-00. D.S.L.M. acknowledges the Kosciuszko Foundation. P.A and F.G acknowledge the FCT through project number UID/04559/2020 (LIBPhys), and contract No. UI/BD/151000/2021.

**Data Availability Statement:** The present data are available on request to the corresponding author.

**Conflicts of Interest:** The authors declare no conflict of interest.

## Abbreviations

The following abbreviations are used in this manuscript:

| | |
|---|---|
| EBIT | Electron Beam Ion Trap |
| DR | Dielectronic recombination |
| TR | Intershell trielectronic recombination |
| QR | Intershell quadruelectronic recombination |

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
