# Peer review of "Hypersatellite Kα Production in Trapped Ar Ions at KK Trielectronic Recombination Energies"

_atoms, doi:10.3390/atoms11030058_

Round 1

Reviewer 2 Report

The authors have provided experimental and theoretical study of the high-order K shell resonances of highly charged argon ions in an EBIT, which are very important for the fundamental research as well as the application in plasma modeling. The paper is well written and structured. I would like to offer the authors the following comments for consideration to revise the manuscript:

1.     Trielectronic recombination (TR) process with electron-ion collision of highly charged ions (HCIs) has already been investigated at the EBIT and storage rings for many different cases, this manuscript shows the evidence of the first observation of KK multishell TR process, they did a lot of simulation work and data analysis, however, since the authors already show the simulated resonance strength of the DR and TR as a function of electron energy, and could you please also show the experimental x-ray spectrum for DR and TR processes as a function of electron energy which already presented in the previously papers related to TR processes at the EBIT (already cited in this manuscript, such as PhysRevA.80.050702, PhysRevLett.107.143201, PhysRevA.88.062706 and so on ). This is extremely important to confirm the evidence of the TR process in this manucript. Otherwise, it is still very difficult to clearly distinguish the high-order recombination process in the present electron-ion collision experiment.

2.     Please show Figure 9, 10 and 11 with the same electron energy range which will be much easier for comparison.

3.     In the conclusion section, please do not cite so many figures, just present the conclusion should be better for the readers to take the information of the paper.

Round 2

Reviewer 1 Report

The manuscript has been improved substantially and most of my comments were answered and taken into account.